# Sex-Based Differences in the In Vitro Digestibility of MCT Emulsions Stabilized by Various Emulsifiers

**DOI:** 10.3390/foods14010131

**Published:** 2025-01-05

**Authors:** Mijal Perez, Carmit Shani Levi, Uri Lesmes

**Affiliations:** Laboratory of Chemistry of Foods and Bioactives, Department of Biotechnology and Food Engineering, Technion–Israel Institute of Technology, Haifa 320001, Israel; mijal.perez@campus.technion.ac.il (M.P.); shanilc@bfe.technion.ac.il (C.S.L.)

**Keywords:** emulsions, sex differences, in vitro digestion, medium-chain triglycerides, protein and non-protein emulsifier

## Abstract

Consumer sex influences phenotypic differences in digestive functions that may underlie variations in food disintegration. This study used an in vitro digestion model to test the hypothesis that emulsions follow distinct digestive pathways in men and women. Model emulsions were prepared using medium-chain triglycerides stabilized by beta-lactoglobulin, alpha-lactalbumin, or lactoferrin, and by three non-protein emulsifiers: Tween 80, lecithin, and sucrose esters. All emulsions were produced by high-pressure homogenization (0.57 MPa, 5 passes) and then subjected to in vitro digestion under simulated conditions of the male or female gastrointestine. Digesta samples were analyzed via confocal microscopy and laser-based particle sizing, revealing that protein-stabilized emulsions were responsive to physiological differences between males and females, whereas emulsions stabilized by non-protein emulsifiers remained mostly unaffected by sex-based differences. Absolute differential analyses of emulsion droplet size-distribution curves showed that changes in breakdown trajectories for emulsions were pronouncedly noticeable in gastric effluents. Further, SDS-PAGE analysis of digesta showed that breakdown patterns of protein-stabilized emulsions are consistent with prior evidence found for healthy adults; however, results under female gut conditions indicated variations in protein clotting that may alter bioaccessible levels of bioactive peptides. Thus, this study underscores the importance of considering consumer biological sex in food design, especially regarding emulsion-based products for targeted digestive responses.

## 1. Introduction

Dietary intake is known to intersect with consumer physiology and inter-human variability on its way to influencing health [1]. In fact, consumer gut physiology is an important underlying determinant of the bioaccessibility and bioavailability of nutrients and other bioactive constituents [2]. To this end, the spatiotemporal variables of the digestive tract, such as gastric acidity, and gastric emptying have been documented to vary along the healthy lifespan and between sexes [1,3,4,5]. Thus, consumer characteristics might be expected to differentially affect the colloidal and biophysical phenomena of food as it passes through the gut. However, consumer sex has largely been neglected in colloidal, nutritional, and even pharmaceutical research, despite numerous indications of its biological relevance, including in relation to digestive functions [3,5,6,7]. At the same time, emulsion digestibility studies have been the focus of lively research that translates basic emulsion digestion research into real-world products and applications [8,9,10]. Such efforts are impactful because lipids are an essential component of the human diet, and their digestive fate is governed by a series of physical and biochemical phenomena occurring mainly in the stomach and small intestine [8,11,12,13].

To date, studies of emulsion digestion have been conducted using both in vivo and in vitro methods [8,14]. In vitro models closely mimic human emulsion digestion, with good correlations with in vivo findings [15]. Such studies highlight the importance of emulsion droplet interfaces, emulsifiers, and various colloidal phenomena (e.g., flocculation and coalescence) as drivers of emulsion digestive behavior [8,11,13,16]. In addition, inter-human variability, such as from infancy to old age, affects gut physiology, which in turn differentially modulates the digestive fate of emulsions [17,18]. In vivo studies further suggest that the complex interactions between food and the digestive system led to different digestive outcomes [1,4,19]. Given these differences, specific protocols have been developed for in vitro simulations of the digestive systems of different populations, such as children, adults, and the elderly [14,17,20]. In fact, an in vivo study even shows that age, sex, and ethnicity influence oral food processing behavior, suggesting possible ethnic differences in digestive outcomes [19]. Regarding sex, documented differences in digestive physiology between men and women are significant enough to warrant the development of sex-specific research on digestion [3,5,6,7,21,22,23]. There is also evidence that the female digestive system undergoes various physiological changes during the menstrual cycle and pregnancy [24,25,26,27,28,29,30,31,32,33]. A review of the scientific literature suggests that there is a paucity of data addressing these differences in emulsion digestion between the sexes and across the lifespan, from infancy to old age [18]. Thus, there is a need for research focusing on differences in lipid digestion between healthy males and females, to facilitate the optimization of food emulsions.

Therefore, this study used in vitro digestion models that mimic the gastrointestinal conditions of males or females to investigate the behavior of a battery of medium-chain triglyceride (MCT) emulsions stabilized with different emulsifiers. It was hypothesized that the differences in digestive conditions and secretions, as well as the composition of the droplet interfaces, alter the trajectory of the emulsions as they pass through the gastrointestinal tract. The aim of this study was to increase our knowledge of the differential digestion of emulsions in males and females, to pave the way for the development of optimized food formulations. Such a deeper understanding of the differences in digestive lipolysis between the sexes could significantly improve and substantiate formulation practices and provide evidence to support public health policies.

## 2. Materials and Methods

### 2.1. Materials

Food-grade bovine lactoferrin (Vivinal Lactoferrin FD, 95.6% protein, Lf) was kindly donated by DMV International (Delhi, NY, USA). Food-grade β-lactoglobulin (BioPURE β-lactoglobulin, 97.6% protein, β-lg) and α-lactalbumin (85% protein, α-lac) were purchased from Xi’an D-Sung Health Biotechnology Co., Ltd. (Hengtian International City, Xi’an, China). TWEEN 80 (T80) was purchased from MERCK. Phosphatidylcholine (Lecithin, Lec) was also purchased from Xi’an D-Sung Health Biotechnology Co., Ltd. Sucrose esters (SP30, E473, Suc) were purchased from SISTERNA, and medium-chain triglycerides (Ph.Eur., USP, MCT) were purchased from Lipoid (Ludwigshafen, Germany). Porcine gastric mucosal pepsin (250 units/mg) and porcine pancreatic pancreatin (8× USP specifications) were purchased from Sigma-Aldrich (Rehovot, Israel). Pepstatin A and α-toluenesulphonyl fluoride (PMSF) were purchased from Sigma-Aldrich (Rehovot, Israel). Fresh bile salt was purchased from the Institute of Animal Research (Kibbutz Lahav, Israel). All chemicals used were of analytical grade, and did not undergo additional purification.

### 2.2. Methods

#### 2.2.1. Preparation of Emulsions

Emulsions were prepared with 1% (*w*/*w*) emulsifier and 20% (*w*/*w*) MCT. The required amount of emulsifier was dissolved in water by stirring for 5 min. This aqueous solution was then mixed with oil to produce a pre-emulsion by vigorous mixing for 1 min using a MICCRA D-1 homogenizer (34,000 rpm). The final emulsion was formed by passing the pre-emulsion five times through a high-pressure homogenizer (Micro DeBEE air-operated, BEE International, Easton, MA, USA) at 0.57 MPa. All emulsions (raw emulsion, E_R_) were stored in a refrigerator and used within 24 h of preparation.

#### 2.2.2. Physical Stability Assessment of Emulsions Using Analytical Centrifugation

The physical stability of emulsions was evaluated using an analytical centrifuge (LUMisizer, L.U.M. GmbH, Berlin, Germany). The analysis involved determination of time- and space-resolved light transmission/extinction profiles (STEP technology) based on analytical centrifugation of four emulsions in triplicate samples, simultaneously. For each sample, 400 µL of emulsion was transferred to a cuvette that was placed in the centrifuge chamber of the LUMisizer. The stability of the emulsions was measured at 25 °C using a wavelength of 865 nm under 2000 rpm for 12.5 h. The data were processed using SEPView 6 software, which quantified the physical destabilization of the samples, expressed as emulsion creaming velocity.

#### 2.2.3. In Vitro Digestion of the Emulsions and Sampling

Acknowledging sex-based differences in digestive functions, emulsions were subjected to in vitro digestion using an autotitrator (Titrando 902, Metrohm, Switzerland) with ‘TIAMO 2.3’ software (Metrohm, Switzerland) programmed to simulate the gastrointestinal functions of healthy females or males. The INFOGEST protocol [34] was followed for male digestion, while a recently described protocol was used for female digestion [22,35]. Table 1 highlights the main differences between the digestion conditions used to replicate female and male digestion.

During digestion, 2 mL of gastric effluent (or 10 mL in the intestinal phase) was aspirated from the reactor at different time intervals: midway through the gastric phase (G_1/2_), at the end of the gastric phase (G_end_), midway through the intestinal phase (I_1/2_), and at the end of the intestinal phase (I_end_). Gastric samples were immediately inactivated with 10 µL/mL Pepstatin A. Intestinal samples were inactivated similarly, but with 14 µL/mL of PMSF.

All digestive effluent samples were mixed and placed on ice before being analyzed fresh for microscopy and laser-based characterization or frozen at −20 °C until further biochemical analysis (as SDS-PAGE). Each type of emulsion was analyzed in triplicate for each digestion model (female or male).

#### 2.2.4. Analysis of Digestive Effluents

Measurement of emulsion droplet size. The droplet size of the emulsions was determined using a static light scattering instrument (Malvern Mastersizer 3000, Malvern Instruments, Worcestershire, UK). A refractive index of 1.33 for the water phase and 1.45 for the oil phase was used for this analysis. Droplet size data were reported in two formats: the volume-weighted mean droplet diameter (D_4,3_ = ∑n_i_d_i_^4^/∑n_i_d_i_^3^, where n_i_ represents the number of droplets of diameter d_i_), which provided a distribution based on droplet volume, and the surface-weighted mean droplet diameter (D_3,2_), which provided a distribution based on droplet surface area. In cases where a two-dimensional comparison is relevant, such as when comparing droplet sizing with microscope images, D_3,2_ data (i.e., analysis of sizes by surface area) have been used as better representatives of droplet circumferences which coincide with the cross-sectional nature of microscopy images. Otherwise, droplet sizing was analyzed by volume of the droplets, i.e., D_4,3_ values, which are a better indicator of droplet mass. Sample volumes varied, according to the digestion phase, to obtain optimal opacity for reliable measurements: less than 100 µL for raw emulsion (E_R_), 200–600 µL for gastric phase samples, and 3–7 mL for intestinal phase samples. All measurements were performed in duplicate, and their reliability was validated according to the instrument software.

ζ-potential measurement. Dynamic light scattering coupled with capillary electrophoresis (Zetasizer Nano ZS series, Malvern Instruments, Worcestershire, UK) was used to determine the electrophoretic mobility of emulsion droplets. Emulsion samples were diluted 1:100 (*v*/*v*) in DDW and then placed in the test cell. After 120 s of equilibration in the instrument, data were collected from at least 5 sequential measurements per sample, and the scattering data were converted to electrokinetic charge values (ζ-potentials) using the Smoluchowski model. All measurements were performed at least in duplicate.

Confocal microscopy. Samples were stained with Nile Red and Fluorescein isothiocyanate (FITC) for analysis by confocal microscopy. For staining, 200 µL of each effluent was mixed with 1 µL of Nile Red solution (1 g/mL in pure ethanol) and 1 µL of FITC solution (1 g/mL in dimethyl sulfoxide). In turn, 10 µL of the stained sample was transferred to a microscope slide and observed by confocal microscopy (Zeiss LSM 880 Confocal Laser Scanning Microscope). Untreated emulsion samples (E_R_) and gastric phase samples (G_1/2_ and G_end_) were observed at 63× magnification, while intestinal phase samples (I_1/2_ and I_end_) were observed at 25× magnification. Ten images per sample were taken at different points on the slide and analyzed using IMAGEJ FIJI 2.14.0 JAVA 1.8.0.

Protein degradation by SDS-PAGE. To evaluate differences in protein degradation under female or male digestive conditions, sodium dodecyl sulphate polyacrylamide gel electrophoresis (SDS-PAGE) was performed in two ways: [i] gel 12% SDS-PAGE and [ii] gel 12% TRICINE (stacking and resolving part of the gel with different concentrations of Acrylamide (40%), buffer (Tris-HCl/DDW, pH = 8.8), DDW, APS and TEMED, according to the protocol). Running conditions were 140 V for 120–140 min in running buffer (Tris-Base/Tricine/SDS/DDW). For Tricine gels, samples (diluted or not in water, as necessary, to obtain the same concentration according to the dilution during digestion analysis) were treated under heating conditions (10 min at 70 °C) with a buffer (Tris-Cl/glycerol/SDS/Coomassie blue R-250/DTT/DDW). Gels obtained after separation were washed in fixative solution (300 mL ethanol, 100 mL acetic acid, and 600 mL DDW), rinsed with DDW, and stained with Coomassie Blue R-250. The gels were then destained with destaining solution (100 mL acetic acid, 200 mL ethanol, 400 mL DDW) for 10 min and placed in 10% acetic acid for 30 min or until bands were clearly visible and good contrast with the background was achieved. All gels were scanned using a Microtek 9800XL Plus scanner (Microtek, Carson, CA, USA).

## 3. Results and Discussion

Emulsion digestion in healthy adults and even infants has been extensively explored [8,14,15,17,18,36]. However, differences in the digestive behavior of emulsions between healthy women and men, arising from physiological differences, have yet to be elucidated, and receive little attention in food research [3,6,7,22]. In this work, the behavior of a battery of MCT emulsions (Figure 1), stabilized by protein or non-protein emulsifiers, was studied under gastric and intestinal conditions. The protein emulsifiers were β-Lactoglobulin (b-Lg), α-Lactalbumin (α-La) and Lactoferrin (Lf) which are major whey proteins and well-studied as emulsifiers [37,38,39]. Their amino acid sequence, MW, isoelectric point and other structural elements are also well-detailed in the literature, which indicates b-Lg and a-La have similar isoelctric points at 4.5–5.0, unlike Lf, whose pI is above 8.0 [39]. Concomitanlty, Tween 80 (T80), Lecithin (Lec) and Sucrose ester (Suc) were used as non-protein emulsifiers, due to their wide application in food systems [40,41,42]. In turn, all emulsifiers were used to prepare test emulsions using high-pressure homogenization, resulting in monodispersed droplet distributions with submicron sizes, except for the lecithin-stabilized emulsions, which displayed larger droplets, predominantly in the 1–100 µm range (Figure 1). Overall, these emulsions were found to be physically stable, with no noticeable changes over two months of storage. This was further supported by measurements of creaming velocity and zeta potential (ζ), which showed absolute values greater than 30 mV (insets in Figure 1), which are considered to be stability indicators for commercial food emulsions [43]. In addition, accelerated stability tests under analytical centrifugation confirmed the high stability of the emulsions, which were found to have calculated separation velocities between 28.4 and 52.0 mm/day at 470 relative centrifugal force (RCF). These values can be equated to 0.06–0.11 mm/day (insets in Figure 1) under normal gravity, which is well below the 3 mm/day threshold considered for shelf-stable food emulsions [43].

### 3.1. Sex-Specific Differences in Emulsion Digestive Behavior

Various emulsions prepared using different emulsifiers were subjected to in vitro digestion under either male- or female-consumer conditions, and were found to exhibit different colloidal behavior, as determined by confocal imaging and laser-based droplet size- distribution curves (Figure 2).

These D_3,2_ colloid size analyses, coupled with direct observations, demonstrated that the β-Lg stabilized emulsion is differentially susceptible to the gastric conditions of males and females. Interestingly, the findings of male digestion are consistent with previous reports on the destabilization of β-Lg or milk protein-stabilized emulsions under gastric conditions in healthy adults [8,10,44,45,46,47]. Thus, the results presented here highlight two points: one is that emulsion destabilization in the gut of males appears to be consistent with numerous observations related to emulsion digestion in healthy adults. The other is that females may have a different digestive trajectory from that commonly reported in the literature. In addition, the T80 stabilized emulsion did not show any noticeable differences in digestion under male- or female-gut conditions or any significant difference from previous studies [10,48]. This trend was also supported by the D_4,3_ droplet size analyses shown in Figure 3.

This further supports differences in droplet size distributions during the digestion of β-Lg stabilized emulsions in males or females. There are marked differences between males and females during gastric digestion of the β-Lg stabilized emulsion, but these appear to be attenuated during the intestinal phase of digestion. Specifically, β-Lg stabilized droplets were found to be prone to increased coalescence under male gastric conditions, which was less evident under female gastric conditions. In the latter, emulsion droplets were found to exhibit more pronounced flocculation. The differences seen in Figure 2 can be attributed to competing mechanisms of emulsion destabilization: coalescence driven by protein emulsifier degradation, and gastric pH-induced droplet flocculation when pH is close to the protein isoelectric point. It appears that coalescence tends to dominate emulsion breakdown in males, whereas flocculation is more prevalent during the gastric phase of female digestion experiments. Thus, this work highlights the fact that consumer sex differences in digetive functions (see relevant details in Table 1) account for differences in gastric pH and residence time, which in turn modulate the kinetic balances between the competing mechanisms of emulsion instability in the stomach.

Concomitantly, no significant differences were observed in the digestion of T80 stabilized emulsions, possibly due to the reduced sensitivity of the non-proteinaceous emulsifier to gastric enzymes and the physiological pH gradients. Thus, Figure 2 and Figure 3 suggest that β-Lg stabilized emulsions have unique transformations, driven by consumer physiology, which are abolished in the case of the non-protein emulsifier T80. Further emulsion formulations were therefore prepared using other protein and non-protein emulsifiers to explore the possibility of different patterns in how stabilizers affect emulsion behavior in the gut of males or females.

### 3.2. Impact of Emulsifier Type on Emulsion Digestion in Males in Females

Based on initial evidence, it was hypothesized that protein-stabilized emulsions would be more sensitive to the digestive physiology of males or females than non-protein-stabilized emulsions. Therefore, two additional protein emulsifiers (alpha-lactalbumin [α-La] and lactoferrin [Lf]) and two non-protein emulsifiers (lecithin [Lec] and sucrose ester [Suc]) were used to prepare emulsions and subjected to male or female in vitro digestion models. To further investigate the colloidal behavior of these emulsions during simulated digestion, particle-size distribution analysis was performed on digestion aspirates, and is summarized in Figure 4.

The results shown in Figure 4 confirm the hypothesis that emulsions stabilized with proteinaceous emulsifiers show significant differences in droplet size distributions during male or female gastric digestion, in terms of droplet distributions and mean droplet population (D_50_, marked as vertical lines in Figure 4). Again, these differences seem to disappear in the intestinal phase, where no major differences between protein emulsifiers or sexes are observed, suggesting that the emulsifier type has its most significant effect during gastric digestion. Non-protein stabilized emulsions, on the other hand, did not show significant sex differences during gastric and intestinal digestion.

To better understand the differences in the droplet distribution curves during gastric and intestinal digestion, the absolute difference between the relevant droplet-size curves was calculated, with plots presented in Figure 5. This integrative analysis highlights the fact that observed sex differences in emulsion droplet distributions are apparent during gastric digestion of protein-stabilized emulsions, diminishing during intestinal digestion. Meanwhile, analysis of non-protein stabilized emulsions showed negligible sex differences. These results suggest that gastric destabilization of protein-stabilized emulsions is influenced by consumer sex physiology. This is likely due to the differential digestive proteolysis of proteins adsorbed at droplet interfaces, competing with pH dynamics that may promote droplet flocculation. Further analysis was therefore carried out to examine protein degradation during digestion.

### 3.3. Sex-Based Differences in Breakdown of Protein Emulsifiers

Recent studies highlight the fact that protein digestion and metabolism may be affected by the sex of the consumer [3,22]. Studies also show that protein adsorption on droplet interfaces can induce allosteric changes affecting protein susceptibility to proteases [45,49]. Therefore, SDS-PAGE analysis was performed on fresh emulsions and gastric effluents of the corresponding protein-stabilized emulsions, with relevant images shown in Figure 6.

The analysis in Figure 6 shows distinct patterns where β-Lg showed some gastric persistence, α-La showed less persistence and Lf was rapidly degraded, consistent with previous observations for gastric digestion of proteins in solution [36,45,50,51,52,53,54]. Interestingly, other reports dealing with whey protein digestion when applied to emulsions also follow a similar general trend [55,56,57,58,59]. There is evidence that protein adsorption at droplet interfaces can increase protein proteolysis, compared to the free soluble form of the proteins, as shown for β-Lg [45]. Furthermore, the disappearance of the protein bands during gastric digestion revealed a subtle sex difference. In fact, for both β-Lg and α-La, the proteins were slightly more persistent in healthy men.

This finding is also consistent with a previous report on the digestion of these proteins in solution [22]. At the same time, both proteins showed a decrease in bioaccessible protein levels in the middle of gastric digestion, which then increased towards the end of this phase. This is likely to be due to protein coagulation and attenuated clot disintegration during gastric digestion, leading to fluctuations in protein and peptide bioaccessibility, a phenomenon reported in both in vitro and in vivo studies [60,61].

Furthermore, recent work has highlighted the notion that subtle differences in the digestive breakdown patterns of dietary proteins may lead to differences in the bioaccessibility of bioactive peptides [22]. Overall, the observed differences underscore the complexity of protein digestion and the importance of considering sex differences in consumer gut physiology. Therefore, further work is needed to elucidate the composition of bioaccessible fractions that may be formed in the gut of males or females and to investigate their biorelevance, nutritional importance and potential health implications.

## 4. Conclusions

This work investigated possible differences in the degradation of emulsions in the gastrointestinal tract of males or females, which has been scantly addressed in food and colloid research. Using relevant in vitro digestion models [22], it was possible to gather evidence showing that protein-stabilized emulsions exhibit differential gastric destabilization under conditions simulating the stomach of males or females. This has been attributed to sex-based differences in gastric functions which tilt the kinetic balance between competing mechanisms of emulsion coalescence and flocculation. Moreover, the observed differences have been postulated to arise from the possible conformational changes that proteins undergo upon adsorption onto droplet interfaces, which delineate their differential susceptibility to clotting and proteolysis under the divergent gastric conditions of males and females. On the contrary, no differences were observed when commercial non-protein emulsifiers such as Tween 80, lecithin or sucrose esters were used in emulsion formulations. Furthermore, the consistency of our findings with previous reports investigating the gastric digestion of emulsions in healthy adults highlights the notion that the digestive fate of foods in the gut of females has not received the needed attention. Recognizing that sex is a fundamental source of phenotypic variability in human gut physiology, further work is needed to highlight differences in the digestive fate of foods. Improving our understanding of the differential digestibility of foods could help to rationally design and optimize foods for optimal bioaccessibility in men and women.

## Figures and Tables

**Figure 1 foods-14-00131-f001:**
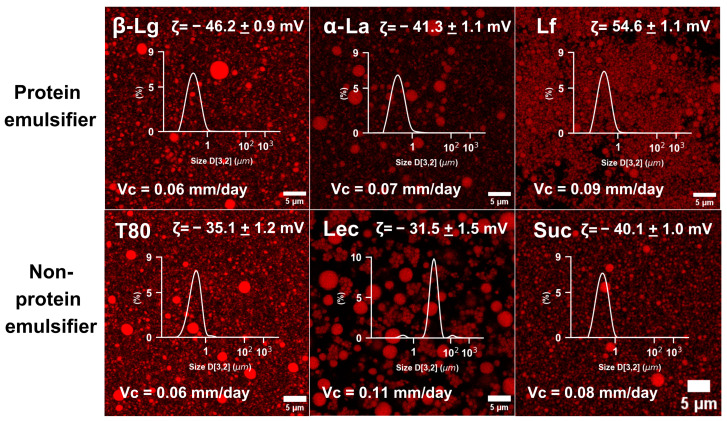
Key colloidal characteristics of emulsions used in this study. Confocal *micrographs* of raw emulsions produced through high-pressure homogenization overlaid with D_3,2_ droplet size-distribution curves, averaged zeta (ζ)-potential parameters (*n* = 10) and average of creaming velocity (Vc) measured after two months with the emulsion stored in the refrigerator during that period (*n* = 3). Micrograph scale bars represent 5 µm. The oily phase has been stained with Red Nile, and appears red.

**Figure 2 foods-14-00131-f002:**
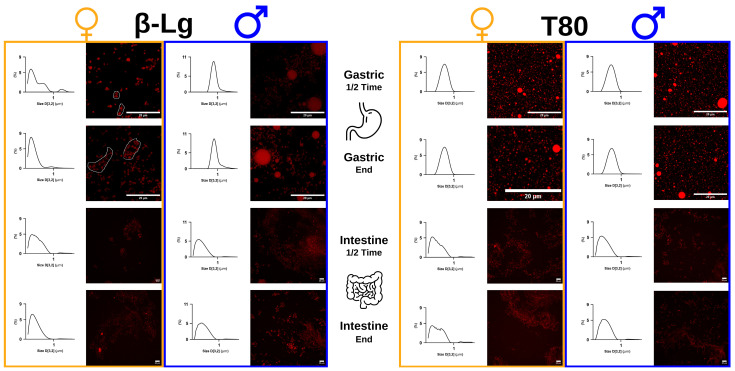
Comparisons of digestive effluents of emulsions stabilized by β-Lg or T80 during female (orange) or male (blue) in vitro digestion. Confocal micrograph of gastric and intestinal samples collected in the middle of the gastric phase (G_1/2_), at the end of this phase (G_end_), at the middle of the intestinal phase (I_1/2_) and at the end of the digestive process (I_end_). Micrograph bar scale represents 20 µm. Next to each micrograph: mean D_3,2_ droplet size-distribution curves (*n* = 15).

**Figure 3 foods-14-00131-f003:**
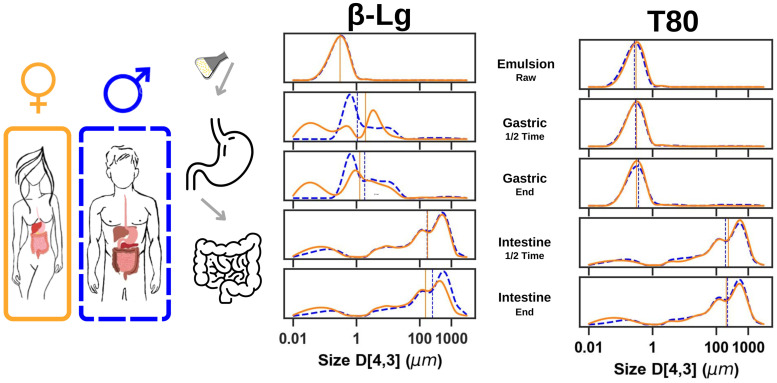
Comparisons of female (orange) and male (blue) in vitro digestion for emulsions stabilized by β-Lg or T80. D_4,3_ droplet size distribution curves of the raw emulsion (E_R_) and all digestive effluents: middle of the gastric phase (G_1/2_), at the end of this phase (G_end_), in the middle of the intestinal phase (I_1/2_) and at the end of the digestive process (I_end_). The mean value (D_50_) for each of the droplet size-distribution curves is shown in solid orange (female) or dashed blue (male) (*n* = 15).

**Figure 4 foods-14-00131-f004:**
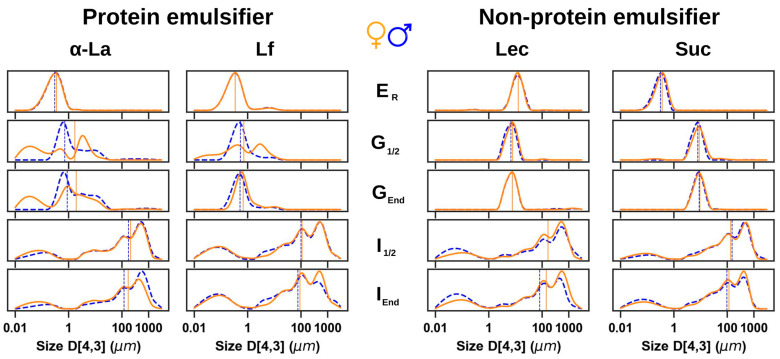
Comparisons of female (orange) and male (blue) in vitro digestion for emulsions stabilized by protein emulsifier (α-Lactalbumin (α-La) or Lactoferrin (Lf)) or non-protein emulsifier (Lecithin (Lec) or Sucrose Ester (Suc)). D_4,3_ droplet size-distribution curves of raw emulsion (E_R_) and all digestive effluents: mid-gastric phase (G_1/2_), at the end of this phase (G_end_), mid-intestinal phase (I_1/2_) and at the end of the digestive process (I_end_). The mean value (D_50_) for each of the droplet size-distribution curves is shown in solid orange (female) or dashed blue (male) (*n* = 15).

**Figure 5 foods-14-00131-f005:**
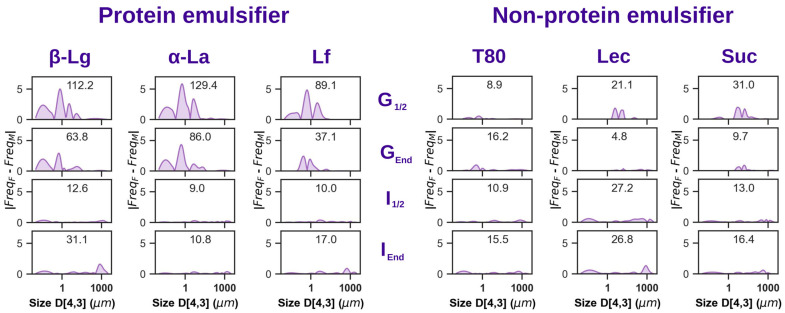
Absolute difference of droplet-size distribution curves found in females and males. Protein emulsifier (β-Lg or α-La or Lf) on the **left** and non-protein (T80 or Lec or Suc) emulsifier on the **right**. The cumulative absolute-difference value (area under the curve) is indicated inside each plot.

**Figure 6 foods-14-00131-f006:**
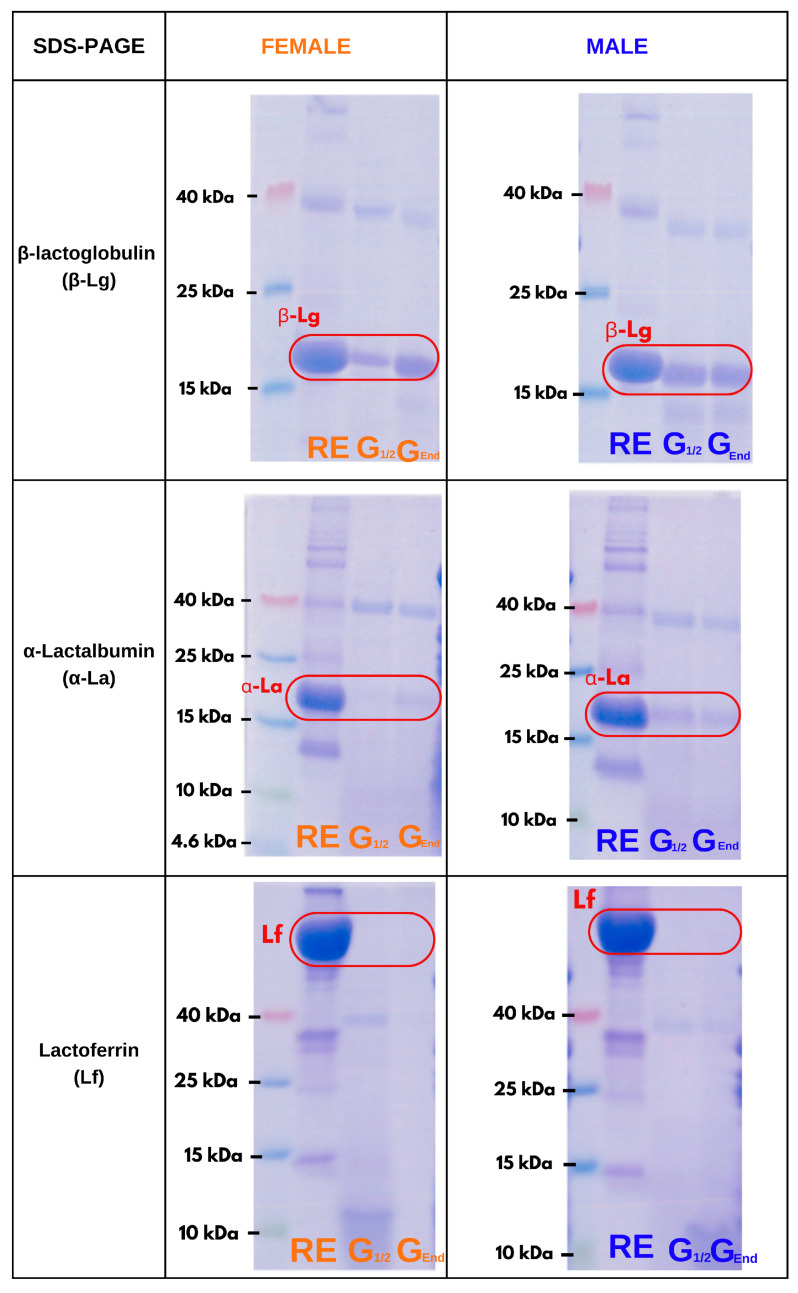
Comparison of SDS-PAGE analysis for raw emulsions and gastric effluents from female (orange) or male (blue) in vitro digestion of emulsions stabilized with β-Lg or α-La or Lf.

**Table 1 foods-14-00131-t001:** Summary of key aspects differentiating in vitro conditions used to simulate digestion in females (left column) or males (right column). Simulated salivary fluid (SSF), simulated gastric fluid (SGF) and simulated duodenal fluid (SDF) were all made from a stock aqueous buffer containing different salts (KCl, KH_2_PO_4_, NaHCO_3_, NaCl, MgCl_2_(H_2_O)_6_, (NH_4_)_2_CO_3_).

Phase	Female	Male
Mouth Emulsion: SSF (1:1)	SSF: NaCl 16.8 mM	SSF: NaCl 13.5 mM
Gastric Oral Bolus: SGF (1:1)	Pepsin 1600 U/mL pH gradient 4.5 to 2.1 Duration: 3 h	Pepsin 2000 U/mL pH gradient 3.2 to 1.5 Duration: 2 h
Intestine Gastric bolus: SDF (1:1) Pancreatin (2000 U/mL)	Bile salts: 10 mM (Taurocholic acid sodium salt: Sodium Glycodeoxycholate) (1:1)	Bile salts: 15 mM (Taurocholic acid sodium salt: Sodium Glycodeoxycholate) (3:2)

## Data Availability

The original contributions presented in this study are included in the article. Further inquiries can be directed to the corresponding author.

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
