# Peer review of "Sex-Based Differences in the In Vitro Digestibility of MCT Emulsions Stabilized by Various Emulsifiers"

_foods, 2025, doi:10.3390/foods14010131_

Round 1
Reviewer 1 Report
Comments and Suggestions for Authors
This work investigated possible differences in degradation of protein and non-protein stabilized emulsions in the gastro and intestinal tract of males or females. The results are very interesting and informational. I recommend it needs revision. Some comments for the authors:
1. Please add the order like a, b, c… in each picture to explain the colloidal behavior in each stage clearly.
2. Please specify all the units at the axis of the droplet size distribution curve in Figure 1 and 2.
3. The size labels in Figure 2 are dim. The droplet size distribution curves may not match the confocal micrograph in Figure 2, please check it.
4. Why do the emulsions stabilized by β-Lg appear to flocculate in females and coalesce in males at the end of the gastric phase? Please discuss in more detail.
5. Have you considered the difference in the concentration of NaCl between females and males with β-Lg stabilized emulsion during the simulated salivary stage?
6. Please add the method of calculating the relevant droplet size curves in Methods.
7. How about add a word of “Simulated” in the title? And clarify this in the abstract.
Author Response
Please see the attached letter of response to reviewers' comments and suggestion

Reviewer 2 Report
Comments and Suggestions for Authors
The manuscript entitled “Sex-Based Differential Digestibility of MCT Emulsions Stabilized by Various Emulsifiers” is an article addressing the interesting issue of differences in digestion of protein-stabilized emulsions and non-protein surfactant stabilized emulsions depending on the gender of the consumer. However, in my opinion, it still requires a significant contribution from the authors in terms of scientific value.
1) The topic of the paper is interesting unfortunately with such a limited number of research methods the results should be intensively discussed with the literature. The authors themselves defined the section as Results rather than Results and Discussion. Discussion was unfortunately scarce there. Among other things, there is a lack of a statement of how the pH conditions were at these stages of digestion and how the protein would behave under such conditions. There is a lack of characteristics of the proteins chosen for the study and an explanation of why such proteins were chosen, whether they were chosen on the basis of similarities or differences in structure. What I miss here is the connection of already existing knowledge with the results of the authors. There is a lack of solid inference and linking of conclusions from the results obtained by different methods.
2) Lines 188-191 should be reformulated
3) Line 194 - Such values means what values? this value of 30 is what limit given by other authors?
3) Editing issues are also important to me. I don't understand why bold text, underlining in descriptions of drawings? Why references are in superscript? Why are the Latin names not in italics (lines 209, 215, 231, 259, 297, 313)? In keywords, the abbreviation MCT should be replaced by the full name. The font in Table 1 should be in accordance with editorial requirements, as well as the caption in figure 4. Line 142 - subscripts are missing. should the unit milliliters be written as ml or mL - once the authors write so, and once differently.
Line 183 - “has been the heart” - is a colloquial term, should be avoided in scientific articles. FIT abbreviation in the caption of Figure 1 - abbreviations in captions should be developed. The description of chapter 3.1 is incorrect - it sounds like a statement. D50 - 50 should be in the subscript.
Author Response

(The authors gave the same response as above.)
